# A Systematic Literature Review of Predictive Maintenance for Defence Fixed-Wing Aircraft Sustainment and Operations

**DOI:** 10.3390/s22187070

**Published:** 2022-09-19

**Authors:** Michael J. Scott, Wim J. C. Verhagen, Marie T. Bieber, Pier Marzocca

**Affiliations:** 1School of Engineering (Aerospace Engineering and Aviation), Royal Melbourne Institute of Technology (RMIT), Melbourne, VIC 3000, Australia; 2Air Transport & Operations, Faculty of Aerospace Engineering, Delft University of Technology, 2629 HS Delft, The Netherlands

**Keywords:** aircraft, decision-making, defence, diagnostics, maintenance, predictive, prognostics, uncertainty

## Abstract

In recent decades, the increased use of sensor technologies, as well as the increase in digitalisation of aircraft sustainment and operations, have enabled capabilities to detect, diagnose, and predict the health of aircraft structures, systems, and components. Predictive maintenance and closely related concepts, such as prognostics and health management (PHM) have attracted increasing attention from a research perspective, encompassing a growing range of original research papers as well as review papers. When considering the latter, several limitations remain, including a lack of research methodology definition, and a lack of review papers on predictive maintenance which focus on military applications within a defence context. This review paper aims to address these gaps by providing a systematic two-stage review of predictive maintenance focused on a defence domain context, with particular focus on the operations and sustainment of fixed-wing defence aircraft. While defence aircraft share similarities with civil aviation platforms, defence aircraft exhibit significant variation in operations and environment and have different performance objectives and constraints. The review utilises a systematic methodology incorporating bibliometric analysis of the considered domain, as well as text processing and clustering of a set of aligned review papers to position the core topics for subsequent discussion. This discussion highlights state-of-the-art applications and associated success factors in predictive maintenance and decision support, followed by an identification of practical and research challenges. The scope is primarily confined to fixed-wing defence aircraft, including legacy and emerging aircraft platforms. It highlights that challenges in predictive maintenance and PHM for researchers and practitioners alike do not necessarily revolve solely on what can be monitored, but also covers how robust decisions can be made with the quality of data available.

## 1. Introduction

Current strategies and policies for aircraft maintenance serve the goal of keeping aircraft in an airworthy state, meaning they are safe for continued operations. Continued airworthiness must be ensured while meeting objectives regarding the availability and cost of aircraft operations, to ensure the economic use of aircraft. Current maintenance programs—both in civil and military contexts—predominantly use “fixed-time-interval and preventive maintenance programs… [which] …can lead to unplanned maintenance activities, comprehensive inspections when no damage is present, or unnecessary replacement of undamaged parts” [1,2]. Over the last decades, the continued increase in the introduction and use of sensor technologies, as well as the increased digitalisation of aircraft operations and support, have opened avenues to monitor, assess, and predict the health of aircraft structures, systems, and components. These activities—typically encapsulated using terms such as predictive maintenance, prognostics and health management (PHM), integrated vehicle health management (IVHM) or aircraft health management (AHM)—feed into a condition-based maintenance (CBM) strategy, which is estimated to provide significant benefits in terms of both cost and time. For instance, a potential benefit of EUR 700 million per year for the European aviation industry alone has been quoted [3], which does not account for other regions, let alone other markets, such as defence sustainment and operations.

Besides promising major economic impact, the areas of predictive maintenance, CBM, PHM, IVHM and closely related concepts have attracted increasing attention from a research perspective. As shown in more detail in Section 2 of this paper, these areas have received a recent surfeit of review papers, as well as original research papers studying various aspects of detection, diagnostics, prognostics and decision support in detail. However, despite this increase in interest, existing research has several major limitations:While recent literature includes about 50 review papers addressing predictive maintenance or similar topics, such as PHM, across various application domains (such as aerospace engineering, mechanical engineering, civil engineering, etc.), virtually all authors **do not define their review methodology**. Little to no attention is paid towards the review technique(s), definitions of scope and use of keywords. Furthermore, limited bibliometric analysis is available in the literature; only a few papers make an effort to analyse the state-of-the-art using quantitative analysis. While the findings of the review papers are typically insightful, the lack of definition of the underlying methodology and non-compliance with PRISMA guidelines [4] makes potential bias of these reviews a major issue.There are **no review papers on predictive maintenance which focus on military applications and a defence context**. As will be shown later, this context provides very specific considerations to be taken into account for all stages of predictive maintenance, including detection, diagnostics, prognostics, and decision-making. Current original research papers in predictive maintenance for military applications are available but have never been subjected to a structured review.Despite several reviews covering the area of predictive maintenance and PHM, **most reviews provide limited insight into how the individual stages of predictive maintenance connect** and integrate with each other, especially when considering the use of prognostics output in decision-making.

This paper contributes to the state-of-the-art by addressing these shortcomings. With reference to the first limitation identified above, it aims to provide a systematic two-stage review of predictive maintenance. To address the second issue described above, this review is focused on the defence domain context, with particular focus on the operations and sustainment of fixed-wing defence aircraft. It applies high-level bibliometric analysis and, uniquely, follows this up with relevance and clustering analysis of recent other review papers via natural language processing algorithms, allowing the identification of clusters of research which subsequently guide the discussion and interpretation of specific elements of the typical predictive maintenance process. To address the third limitation, this paper considers these typical elements in a consistent chain spanning from data acquisition to detection, diagnostics, and prognostics of maintenance events (such as failures), followed by decision-making. 

The structure of this review paper is as follows. In Section 2, the review methodology is described in detail, and quantitative results are provided. This includes a high-level bibliometric analysis of predictive maintenance, showing trends in the state-of-the-art over time. Subsequently, a systematic analysis of a set of 50 review papers is conducted using natural language processing techniques, allowing for the identification of particular terms of relevance as well as major clusters of research in predictive maintenance. These results are used to inform the subsequent structure of this paper. The major stages of predictive maintenance are adhered to, leading to a discussion of core aspects of predictive maintenance in defence sustainment and operations in Section 3. At this point, it should be made clear that sustainment in this context refers to the on-going maintenance, system replacement or modification, as well as maintenance programming, resourcing capabilities and strategic planning that ensures defence aircraft are mission ready [5]. Subsequently, major elements of decision-making are discussed in Section 4. This is followed by a synthesis of major challenges and opportunities in predictive maintenance for defence sustainment and operations in Section 5, followed by some brief conclusions.

## 2. Review Methodology: A Systematic Review Incorporating a Bibliometric Approach 

To address one of the identified issues with existing review papers—as mentioned in the introduction—it is imperative to conduct a systematic review while clearly explicating the review methodology. As such, this section supports the overall problem statement of the paper and explains the underlying review methodology. It is important that gaps and challenges in the state-of-the-art are identified as well as trends in progressing prognostics in aircraft maintenance. Therefore, a two-stage review methodology has been implemented, incorporating:**Bibliometric analysis of literature within the scope of application:** the aim of the bibliometric analysis is to present a macro-view of predictive maintenance and its trends over time.**Systematic analysis of existing review papers using natural language processing:** within the relevant body of research, existing review papers on predictive maintenance and closely related terms, such as condition-based maintenance (CBM), integrated vehicle health management (IVHM), Integrated System Health Management (ISHM) and prognostics and health management (PHM), have been collated and analysed using two natural language processing techniques. The aim of this analysis is to identify key clusters of research. To this end, fifty recent review publications have been analysed. The review papers are within the past 20 years, dating from 2000 to 2022. Machine learning and natural language processing algorithms, namely K-means clustering and TF-IDF (Term Frequency-Inverse Document Frequency), have been employed to analyse large amounts of text across the considered fifty documents. These algorithms have been implemented using programming scripts within the scikit-learn Python libraries [6]. Analysis has been performed across all fifty documents to summarise the content into respective clusters of terms denoting the overall focus of each individual paper. The results are shown in the subsequent sections.

### 2.1. Publication Trends in the Relevant Body of Knowledge

In Figure 1, four charts present a macro-view of the current body of knowledge relating to engineering disciplines and specific to terms falling under predictive maintenance. This is contrasted with research data on an alternative maintenance strategy, preventive maintenance, for context. The data are sourced from Scopus (https://www.scopus.com/, accessed in the period of 18 August 2022, one of the largest databases of scientific literature, providing a curated dataset focussed on publications relevant to state-of-the-art research. Analysis of the chart trendlines shows that over the past twenty years, there is a large increase in the number of publications including the terms ‘predictive’ and ‘maintenance’, while in contrast, publication numbers for ‘preventative’ or ‘preventive’ and ‘maintenance’ have not grown substantially.

The publications year-on-year with a moving average of five years to smooth out the variability in publishing cycles are presented in Figure 2. It is evident that there is a large rise year-on-year in predictive maintenance and prognostics publications, compared to a steady flattening in preventative maintenance over the past five years.

Over the previous five years, publications have increased by 19% year-on-year relevant to “Prognostics”, while in the same period, “Diagnostics” related publications have increased by 8.1% year-on-year. This is indicative of the rising shift towards prognostics capabilities and demand in engineering applications. Similarly, over the past five years, publications have increased by 30.4% year-on-year relevant to “Predictive Maintenance”, contrasting this with a lower year-on-year increase of 4.8% relevant to “Preventative OR Preventive AND Maintenance” publications. At a high-level this is a clear distinguishing trend away from reactive preventative maintenance practices and a move towards more proactive predictive maintenance.

### 2.2. TF-IDF and K-Means Clustering of Review Papers

This section takes a more in-depth view of the state-of-the-art, narrowing the analysis to fifty review papers. These papers have been manually selected for relevance to the domain of aircraft predictive maintenance through an initial selection by keyword, subsequent abstract review, followed by full paper review, bringing down the sample from several hundred papers to the selected fifty. This down-selection was performed systematically through identification of papers published within the past twenty years, focusing on review papers only. Furthermore, these review papers were only selected if they had relevance to aircraft maintenance or defence relevance in the context of fixed-wing aircraft, in line with the scope of this review. The relevance was established by a thorough review of individual review papers, including their stated scope, aims and objectives, and application domain(s). Using TF-IDF and unsupervised learning K-means clustering, analysis has been performed to identify and cluster the most critical key terms in the down-selected fifty papers. It is important that the reader understands the basic underlying approach to the TF-IDF and K-means analysis, both for contextualising the authors’ approach and for reproducibility purposes. Hence, key analytical elements are detailed here, including the equations derived in Python library scikit-learn scripts, namely the *TF-IDF* Vectorizer [7]. The term frequency (*TF*) equation is defined by the number of times a term appears in a document, divided by the total number of terms within the document:(1)$$TF\left(t,d\right)=\frac{f_{t,d}}{\sum_{t^{\prime }\in d}f_{t^{\prime },d}}$$
where f is the frequency of the term t within the document d. Similarly, the Inverse Document Frequency (*IDF*) equation is defined as the natural log of the total number of documents N in the corpus, divided by the number of documents D that contain the term t:(2)$$IDF\left(t,D\right)=\log\left(\frac{1+N}{1+f_{t,D}}\right)+1$$

Noting that 1 is added to the variables to ensure no division-by-zero; for example, if the term is not in the corpus. Finally, a TF-IDF score is calculated by multiplying Equations (1) and (2):(3)TF-IDF=TF(t,d)×IDF(t,D)

The TF-IDF score is a simple straight-forward numerical statistic, indicative of how important a word (or term) is to a document in a collection of documents (or corpus). It should be noted that a level of pre-processing of the documents is carried out to remove artefacts of text processing and to avoid terms such as ‘the’, being the most common word in written English language.

The initial results of the TF-IDF analysis are shown in Table 1, where important terms have been systematically identified across the corpus of 50 review publications relevant to predictive maintenance in the engineering disciplines. This analysis is an objective methodology to assess high-level indicators in the state-of-the-art and categorise the documents that share similarities. Furthermore, to identify overall trends, the terms across the corpus can be ranked in order from highest to lowest according to their TF-IDF scores.

This is shown in Table 1, identifying that over the past approximately twenty years, there is a clear trend towards terms such as PHM (prognostics and health management), Fault, Learning (relating to machine learning and deep learning), RUL (remaining useful life) and SHM (Structural Health Monitoring). Plotting the top five terms across the corpus timeline, as shown Figure 3, it is indicative that there is a sustained focus on prognostics over the years. Furthermore, it is evident that there is an increase in machine learning related terms in recent years, while RUL has reduced in importance. However, terms further down the ranking show an emergence of terms such as ‘decision-making’, ‘composite’ and ‘reasoning’. This becomes clearer in the clustering of these terms, which is performed in the subsequent analysis.

TF-IDF is calculated for all fifty review papers over the past twenty years relevant to aircraft predictive maintenance. Further to this, K-means clustering of terms is used to find papers similar in terms to each other. This is an iterative approach, performed by dividing TF-IDF values into K clusters (K denoting a random constant, in this case, the number of clusters). The approach is based on minimising the variance between values, hence ’means’ referring to the averaging of data across the clusters. For reproducibility purposes, the key parameters used in the TF-IDF and K-means analyses are as detailed in Table 2:

The TF-IDF analysis key parameters include limiting terms to only single whole words, excluding terms made up of two or more words. To clarify this, only unigrams have been considered, where the term contains only a single word in sequence. For example, the TF-IDF analysis returned the unigram terms “PHM” and “prognostics”, in contrast to the potential for n-grams, such as “prognostics health management”. Furthermore, a library of common English stop words is used to exclude terms that are insignificant, such as articles, pronouns, prepositions, conjunctions, and additional terms that are artefacts of the document processing, such as “http” and “doi”. The TF-IDF analysis is constrained to 100 terms, so sufficient comparison within the corpus can be made, and manual fine-tuning of parameters is made easier with a review of these mid to high-ranking terms. The key parameters “max_df” and “min_df” ignore corpus terms with maximum and minimum thresholds, respectively, which subsequently decreases processing time. Parameter “max_df” is set to 0.8, which ignores terms that appear in more than 80% of the documents, in turn removing terms that appear too frequently specific to the corpus. Similarly, parameter “min_df” is set to 5, which ignores terms that appear in less than five documents, which removes terms that appear too infrequently.

Using the K-means clustering machine learning algorithm [8], five clusters are set, with key parameter “n_clusters”, a simple nominal amount for ease of analysis, and for reproducibility, a “random_state” parameter of zero was selected to ensure deterministic results. It should be noted that the randomisation parameter was checked for multiple distinct random seeds, and the results were found to be stable. Furthermore, as shown in Table 2, the key parameter “max_iter” was set to 10,000 to ensure consistency across runs. Furthermore, “n_init” was default at ten runs for varying centroid seeds and “init” set to “K-means++” was also a default parameter with faster convergence, compared with random initialisation of centroids.

As shown in Figure 4, the individual review papers are plotted within their respective clusters (A–E) along the vertical axis, and along the horizontal axis against the normalised TF-IDF score for each paper. The reader can use Table 3 to refer to the fifty papers, which are referenced in Figure 4. It should be noted that the normalised TF-IDF score is indexed for each paper from the summation of the top 100 terms, and this min–max normalisation is described by Equation (4) as follows:(4)$$Normalised\;TF\text{-}IDF\;score=\frac{TF\text{-}IDF_{o}-TF\text{-}IDF_{min}}{TF\text{-}IDF_{max}-TF\text{-}IDF_{min}}$$

In Figure 4, the normalised sum TF-IDF score per paper shows to what degree individual terms within a paper vary relative to their cluster.

In clusters A and B, there is only one paper per cluster, meaning they are significantly different in comparison to the corpus. While in clusters C, D and E, most of the papers are grouped together with greater similarity and can be briefly summarised with the high-level terms as shown in Table 3. The horizontal axis in Figure 4 shows the variability within each cluster and how much each paper differs in terms. For example, review paper 27 in cluster E differs from the cluster with a relatively high TF-IDF score, and this is a result of the content being comparable in-depth on RUL and degradation, but specific to machine bearings. This highlights how a review paper within the overall corpus and subsequent cluster can differentiate with greater depth on a specific aspect of predictive maintenance.

In Table 3, the top ten terms belonging to clusters A through to E are shown with the corresponding fifty review papers referenced. Although this is a convenient snapshot of the papers, distilled down to ten terms, only limited conclusions can be drawn from each cluster, as there is overlap in cluster terms. As such, the sections to follow provide greater depth of analysis relative to several terms of particular relevance in the identified clusters.

In summary, this section helped frame the literature review problem statement. With the support from the bibliometric methodology, this literature review identifies what is the current focus of the body of knowledge, as expressed in clusters of relevant terms. The focus of the remainder of this review paper is on various terms identified in the clusters, while ensuring connection with the high-level research gaps identified in the introduction and problem statement. As such, the next sections aim to review the existing literature relating to predictive maintenance and decision-making for defence aircraft sustainment and operations.

## 3. Predictive Maintenance in Defence Sustainment and Operations

Aircraft structural PHM is progressing towards proactive condition-based maintenance technology. Structural PHM aims to provide actionable data-driven decision-making, greater operational efficiencies, reduced maintenance costs and improved structural design [43]. Although the basis and notion has been in development for several decades with various forms of SHM systems, it has struggled to be fully implemented effectively on in-service aircraft [57]. This is due in part to business processes [58], high sustainment costs, data analysis capabilities, data ownership rights and computer technology limitations [53,54]. A recent review of integrated vehicle health monitoring systems highlights the progress to date and the future challenges in adopting this technology, as well as the human and regulatory hurdles [50,59]. Moreover, the regulations of implementing condition-based monitoring, with the growing use of on-board condition monitoring systems, require guidelines to reach industry standards [60], and be rigorously tested before likely wide-spread operator adoption is realised. This is a slow process for both civil and military aircraft operators [61]. At present, standards are available for SHM technologies on fixed-wing aircraft, but this is restricted to civil aerospace applications [62].

For ease of reference for readers, Table 4 provides a brief list of relevant research into predictive maintenance, including the type of paper, the year of publication and a brief description of the paper contribution(s) and relevance.

An in-depth discussion of predictive maintenance for defence sustainment and operations is provided in the following five subsections. 

First, the current landscape of sustainment and operations is briefly outlined in Section 3.1. One of the major issues within this context is the issue of unscheduled maintenance; several studies highlighting unscheduled maintenance occurrences in a defence context are discussed in Section 3.2. Prevention of unscheduled maintenance is one of the primary drivers behind the adoption of sensor technology and subsequent Structural Health Monitoring (SHM) approaches; as such, SHM for defence fixed-wing aircraft is discussed next in Section 3.3. Two major phases of SHM are on-board sensor diagnostics and prognostics, which are the subject of the final two subsections (3.4 and 3.5). Preceding this discussion, it is useful to outline that the typical flow for data-driven SPHM systems is discussed in [35,63]. In most cases, SPHM begins with the on-board architecture available for data acquisition and condition monitoring systems, which is followed in most cases by off-board data processing, feature extraction and analysis to be used for statistical modelling and fault diagnostics. Subsequently, prognostics algorithms may be developed, tested, and applied. To distinguish the step from diagnostics to prognostics, [52] proposes a simple delineation: diagnostics involves identifying and quantifying the damage that has occurred (and is thus retrospective in nature), while prognostics is concerned with trying to predict the damage that is yet to occur. Although diagnostics may provide useful business outputs on its own, prognostics often relies on diagnostic outputs (e.g., fault indicators, degradation rates, etc.) and therefore cannot be viewed in isolation. 

### 3.1. Current Sustainment and Operations

Modern aircraft operator maintenance programs are underpinned by the regulatory and industry guidelines of Reliability-Centred Maintenance (RCM) processes [64] and the current Maintenance Steering Group (MSG-3) [65], based on the iterated approach first developed for the Boeing 747-100 in 1968. The first approach, MSG-1, was primarily a preventative maintenance program addressing operational safety and identifying hidden functional failures through interval driven, hard-time limited and on-condition inspection tasks. The updated approach, MSG-2, in 1979 further developed processes, adding condition-based maintenance practices that monitor aircraft systems at a component level, though such a bottom-up approach has a greater economic burden with an increase in tasks. The latest maintenance program process, MSG-3, takes a top-down approach, which considers the impact a system failure has to operational cost and safety if scheduled maintenance is not performed, which in turn reduces the number of tasks and increases aircraft availability [54]. Military aircraft are typically designed on a Damage Tolerance Analysis (DTA) principle, which for aircraft sustainment is a deterministic fail-safe approach [66,67]. Military aircraft accrue greater structural damaging flight hours compared to passenger aircraft, experiencing higher normal load factors with higher cumulative occurrences in varying operational conditions [68]. A fail-safe approach does not consider individual aircraft conditions, hence the practice of applying scatter factors and safety factors to a maintenance program. This practice has a tendency for overly conservative life estimates and inspection intervals, which leads to inefficiency in a maintenance program. For this reason, the introduction of condition-based maintenance with the support of SPHM technologies has the potential to lower the costs of inspections and availability can be increased, including the life-time of the structure [69].

### 3.2. Unscheduled Maintenance Events

Aircraft component failure rates over time, shown in [70], identify that preventative maintenance practices are inadequate in addressing unscheduled maintenance. Indeed, an analysis of United Airlines aircraft maintenance demonstrated that 89% of components exhibited no wear-out. This led to the conclusion that the performance of the component could not be improved by setting age limits. Failure rate and type for mechanical systems on aircraft can be characterised by the following two groups—age-related and random, accounting for 11% and 89%, respectively, of failure types. Age-related failures are characterised as either occurring in a ‘bathtub’ failure curve (4%), wear-out (2%) or fatigue (5%). Random failures are the majority of failure rate types, being break-in period (7%), random (14%) or infant mortality (68%). It was found that conditional probability of failure at a period remained constant and reliability would remain constant or actually improve with age. This challenges conventional maintenance regimes and brings significance to condition-based or predictive maintenance in identifying patterns as a function of the condition rather than the age of components and wear-out rates. This is found in military technical manuals used for aircraft operations, which again state that 89% of system failures cannot benefit from simple rules that limit operating age [71]. As such, the procedures for assessing the risk of aircraft system failures are a mixture of quantitative and qualitative inputs, determined analytically using reliability data and by engineers subjectively ranking severity with discrete occurrence levels. Similarly, this approach is common in other domains, such as maritime vessel system failure behaviour; in [72], it is found that failures could be reduced with a predictive maintenance strategy. This reinforces the need for SPHM to deliver higher-resolution information for data-driven and quantitative decision-making for fleet managers—see discussion in Section 4.

### 3.3. Structural Health Monitoring in Military Fixed-Wing Aircraft

Developments in SPHM systems for next generation military aircraft have seen a shift from rudimentary sensor data monitoring to end-to-end integrated asset management. An example of this is implemented for the Lockheed Martin F-35 Lightning II Joint Strike Fighter, which has information-rich systems supporting a mandated Autonomic Logistics Information System (ALIS) [73]. The future of PHM systems in isolating faults, preventing spurious faults (or “Fault not Found”, as opposed to NFF (No Fault Found)), considering that studied unscheduled removals of LRUs (Line-Replaceable Units) can be in excess of 40% No Fault Founds [74]. Progressing towards more condition-based maintenance approaches is shown in the integration into the latest next-generation military aircraft, for the purpose of combining on-board sensory systems with off-board support architecture [75]. The SPHM system capabilities are detailed in [76], introducing a data-driven approach to supporting the F-35 fighter aircraft and highlighting—now over a decade ago—the vision for enhanced diagnostics, prognostics of material condition and prediction of remaining useful life and time to failure of components. This includes aircraft health management to provide decision-making support to optimise planning or defer maintenance and manage the remaining life of components. However, the article demonstrates that the PHM architecture is centred on aircraft component systems and the short- to medium-term diagnosis of faults, as opposed to the airframe structures, which typically have longer intervals in fault occurrences. Furthermore, the example provided in [76] highlights the significant effort involved in off-board data analysis and the development of prognostic algorithms required to support fleet management decision-making, which is the scope of the subsequent section.

A recent condition-based maintenance (CBM) technology impact study [77], published by the Australian Defence Science and Technology Group (DSTG), found that effective use of CBM also requires establishing a data management strategy, analysis, decision support, as well as development of prognostic and diagnostic algorithms. Furthermore, the study highlights past examples of CBM in the air domain, which have needed to incorporate allowances for the short life cycle of certain CBM technology. These are challenged by the associated rapid obsolescence rates of parts, in the case of extended planned withdrawal dates of fleets, and unclear contractual arrangements for data ownership and data mining from CBM systems. Recent technical reports and United States Department of Defense (DoD) orders have mandated CBM, as emphasised by DoD Instruction 4151.22 [78]. In such case, CBM is expanded and termed CBM+, which shall be used as the principal consideration for the selection of proper maintenance concepts. CBM+ is defined as the application and integration of processes, technologies, and knowledge-based capabilities to achieve target availability, reliability, and operation and support costs of Marine Corps’ systems and components across their lifecycle [79]. This development is further highlighted in Figure 5.

### 3.4. Diagnostics Approaches for Military Fixed-Wing Aircraft Applications

Aircraft structural anomaly diagnostics has been an area of research for decades, with a focus on damage detection of material components, fatigue, environmental or accidental damage. As mentioned previously, [52] proposes that diagnostics involves identifying and quantifying the damage that has occurred (and is thus retrospective in nature). Nevertheless, this retrospective understanding may still help to refine existing maintenance policies, helping to decrease unscheduled maintenance events and/or their impact.

Research mainly consists of experimental work carried out on-ground with test coupons using guided wave, lamb waves, Fibre Bragg Grating (FBG) optic sensors, piezo-electric transducers and other experimental SHM technology [81,82,83,84,85,86,87]. Although these approaches tend to deliver consistent and accurate results, it comes at a large cost to maintain sensor systems, modify existing components and time in acquiring data from systems that require skilled labour, and predominately at technology readiness levels (TRL) 5–6, i.e., below implementation level, making it difficult to implement on-wing [13].

A large body of research exists relative to diagnostics. In this review, two categories of diagnostics are discussed in more detail to highlight typical considerations and outputs. 

The first category focuses on identifying non-linearities in aircraft structures using PHM system data. Where structural non-linear behaviour can occur routinely and accumulate over the operational life of aircraft, such non-linear behaviour can impact the structural health of the system and limit aircraft flight envelope performance, for example, Limit Cycle Oscillations (LCO) [88]. Aircraft control surfaces with structural non-linearities could include free-play, a dead-zone in an actuator mechanism and/or loosened mechanical connections. The impact of control surface free-play faults, in the context of SPHM and operations, can be considered high risk, with most aircraft system flight controls ATA Chapter 27 [89] constrained to “Go/No-Go” operational decisions [90]. Furthermore, aircraft manoeuvrability and flight envelope performance, especially relevant to military fighter aircraft, is greatly impacted by excessive free-play in control surfaces, as a function of aeroelastic instabilities [91]. Free-play severity and operational impact vary across platforms; in some instances, small amounts of free-play are not a problem, as the control surface is sufficiently aerodynamically loaded as described in [92] for the horizontal tail of the General Dynamics/Lockheed Martin F-16 Fighting Falcon. However, military specification (MIL-SPEC) MIL-A-8870, based on the wind-tunnel results conducted by the Wright Air Development Centre in 1955 [93], requires free-play for all-movable control surfaces to be within 0.034 degrees (peak-to-peak value) to ensure freedom from LCO during normal operations. The United States Federal Aviation Administration (FAA) adopts the same free-play specification for commercial aircraft. This requirement is typically difficult to meet, and in the case of the F-16 all-movable horizontal tail, it can operate at six times above the allowable free-play angle. Similar control surface free-play studies on LCO characteristics [94,95] suggest that an improved standard for free-play limits could have a substantial impact on future aircraft structural design and sustainment. Considering this current operational constraint, a method to identify non-linear behaviour regions is based on Higher-Order Spectra (HOS) analysis, a signal processing technique revealing non-linear phase-coupling between frequencies within the aeroelastic system, originating from non-linear structural and aerodynamic mechanisms. Bi-spectral and tri-spectral density analysis can be used to identify the quadratic and cubic processes within the system, generated under different flight conditions. This is critical to understanding the transition from linear to non-linear aeroelastic behaviour, including LCO phenomena [96,97]. Recent empirical research using flight test datasets on a military aircraft all-movable horizontal tail has produced accurate and consistent results for such an approach [98,99,100,101]. The framework includes data-driven signal processing system identification, utilising HOS to detect and localise non-linear phase-coupling between frequency components from time-domain sensor data within an aeroelastic system [102,103,104,105]. It is a well-established approach to signal processing, and in early iterations of the technique was used for methods in fatigue crack detection, being much less affected by noise interference, making use of sideband peak counts [106,107]. However, it relies on the manual classification of normalised bi-spectral and tri-spectral density plots to identify quadratic and cubic processes, requiring both high levels of free-play and a good understanding of the platform configuration for accurate results. In order to make use of the diagnostics, a method to quantify the magnitude of free-play is achieved using an Empirical Mode Decomposition (EMD) approach, to sift and extract the non-linear frequencies in the form of Intrinsic Mode Functions (IMF), which is used to directly estimate the free-play magnitude [108]. This framework is a major contribution towards detecting, localising, and quantifying the free-play magnitude, although manual filtering steps throughout the process limit the tuned framework to a specific platform configuration and flight test sensors, such as accelerometers. The scope of future work has the potential for utilising available on-board sensors (e.g., strains) present on fleet aircraft. This is described in [109], where in the case of strain sensors being the only available channel output, complexity is added in the pre-processing stage. To convert strains into response information that is useful for free-play diagnosis under the data-driven framework, a pre-processing step would involve data-manipulation and data-fusion strategies that may include a modal expansion or virtually expanded sensor data [110], transfer functions, regression or machine-based learning processes, among others. This is critical to developing the next phase of prognostics, understanding the failure mechanisms, and developing the degradation models with the right inputs to estimate the RUL.

The second category of diagnostics considers vibration analysis of aircraft structures, consisting of both composite and metallic materials. Vibration analysis can be used to diagnose global and local damage and is commonly performed by evaluating the variations in natural frequencies and modal shapes. This is an established approach with model-based simulations generating good results with low computational intensive approaches [111], including using artificial neural networks (ANN) techniques [112]. While detection has reasonable accuracy and the location of damage can be identified efficiently, it often stops there, with a subsequent recommendation that this could be used in future approaches combined with condition monitoring systems. Recent experimental testing on an Airbus A350-900 aircraft showed the next steps towards implementation with promising performance [113], where six bi-axial accelerometers attached to the fuselage could localise impact damage with a mean error of two metres or approximately four percent of the effective structure length.

Military fixed-wing aircraft sustainment has a long history of SHM technologies; however, unlike the previous civil aviation example, military aircraft relate largely to loads monitoring for airframe fatigue life analysis and accelerometers measuring load exceedances over longer periods of time [114]. Military aircraft, such as the Panavia Tornado fighter jet and Eurofighter Typhoon, had operational loads monitoring (OLM) technologies at 16 critical load locations. Furthermore, in the case where aircraft are assessed for planned withdrawal date (PWD) extensions, this is common as defence operators fly the aircraft varied to the design conditions or often go beyond the designed life of assets either constrained by budgetary pressures or awaiting fleet replacements. A recent example of this can be found for the Pilatus PC-9/A aircraft operated by the Royal Australian Air Force (RAAF), where an OLM program included placing accelerometers and strain gauges across the aft fuselage and empennage [115]. This was critical to meeting the extended PWD and ensuring it met the sustainment practices aligned with the Aircraft Structural Integrity Program (ASIP) [116]. It remains that strain gauges are the most prevalent sensor available on-board fixed-wing aircraft SHM systems. Military aircraft, such as the A400M and F-18, have trialled acoustic emission technologies and damage monitoring and diagnostics systems; however, this is more specifically used for fatigue damage analysis or crack propagation over longer periods of time. The C-130 transporter and KC-30 tanker transporter have similar OLM systems [114]. 

Fundamental to these on-board SHM technologies is the off-board analysis, which is carried out using signal processing techniques underpinned by statistical methods. Two time-domain statistical parameters proved good diagnostic techniques for anomalies in ball bearings [117]; namely, root mean square (RMS) measuring mechanical and environmental noise levels, which is compared to nominal sensor conditions, coupled with monitoring high Kurtosis (standardised fourth central moment) values measuring vibration data for an increase in spikes, which is symptomatic of faults. A fundamental advantage of data-driven techniques is that the physical system they model does not need to be related, and only needs to relate to the input data, independent of the type of sensor [118]. Machine learning approaches are typically data-driven techniques, which involve a training step initially and a testing step. These machine learning techniques are known as “supervised” learning, while “unsupervised” learning techniques are typically ANN approaches. Once supervised machine learning techniques are trained, they do not require high computational resources and return fast classifications of new input data. For rapid diagnostic frameworks, the speed of classification could be of relevance, while longer lead time prognostics could benefit from either machine learning approach.

### 3.5. Prognostics and Remaining Useful Life (RUL) Prediction for Military Fixed-Wing Aircraft

Prognostics in aerospace structures is often referred to as the early detection of faults, and the capability to predict the progression of a fault condition. Although prognostics is not new to aerospace, it is yet to be fully implemented on a fleet-wide level and varies in application across aircraft structures and systems. The development of prognostics for complex aerospace structures requires a multi-disciplinary approach, a fusion of prognostic methods, exploiting the strengths of various tools to estimate RUL [119]. The estimation of RUL is central to PHM and condition-based maintenance of operational assets, broadly defined in [32]. The RUL of an asset is a random variable and dependent on current age, the operational context and observed condition monitoring. Defining $x_{t}$ as the random variable of the RUL at time t (usage), with the probability density function (PDF) of $x_{t}$ conditional on $Y_{t}$ is denoted as $f\left(x_{t}|Y_{t}\right)$, where $Y_{t}$ is the historical condition monitoring profile up to t. If $Y_{t}$ is unavailable the estimation of $f\left(x_{t}|Y_{t}\right)$ is:(5)$$f\left(x_{t}|Y_{t}\right)=f\left(x_{t}\right)=\frac{f\left(t+x_{t}\right)}{R\left(t\right)}$$
where $f\left(t+x_{t}\right)$ is the PDF of the life at $t+x_{t}$ and R(t) is the survival function at t. The availability of $Y_{t}$ provides more information about the RUL. However, it is a non-trivial task to incorporate $Y_{t}$ into the estimation of $X_{t}$. RUL estimation can be greatly varying in their inputs, and Equation (5) may require more condition monitoring inputs and a data fusion approach to correlate the varying inputs [32]. As such, Equation (5) is not conclusively an effective means of estimating RUL for all systems. Additionally, the matter of how subjective human judgemental information (heuristics) is processed into RUL is an area of research to be addressed and in the scope of decision-making [48]. Condition monitoring enables early detection through diagnostics specific to a system component, but the importance of failure mode predictability in calculating a reliable RUL is critical for scheduling maintenance effectively [120]. 

Several reviews and original research papers consider applications and challenges for prognostics in more detail. Kan et al. [44] assess the suitability of prognostic techniques for non-stationary and non-linear rotating systems, identified by machine type (e.g., aircraft), where vibration data are accessible. The review finds particle filtering (also referred to as sequential Monte Carlo methods using Bayesian inference) performs well for prognostics that require low computational loads for resampling and have a reduced number of samples for approximating future states. This approach applied to crack faults generates satisfactory results as proposed in [121]. Low cost, efficient data-driven approaches can produce very good results, as in a particle filter prognostics approach used for estimating the RUL for battery life as studied in [35]. Given a test degradation curve for battery capacity, Bayesian filtering is carried out to produce a probability distribution function (PDF) at each time stage; as usage time progresses, the PDF narrows at the later battery life stage. Furthermore, prognostics for aircraft brake wear demonstrate the effectiveness of Bayesian Networks for the remaining useful life of brake material thickness, benchmarked against standard degradation or simple extrapolation approaches [122]. Deep learning (DL) in prognostics and health management applications is reviewed in a comprehensive evaluation of current developments over the last decade in [42], remarking on both the potential and challenges. For PHM systems, unsupervised learning techniques are relevant where the true end of life is often unknown, which is often the case for current preventative maintenance practices being carried out, where limited data are available, in part due to cost or impracticalities to label what is a true system health. In most PHM condition monitoring, unsupervised learning approaches, such as signal reconstruction (residual-based) used with data-driven or physics-based approaches can distinguish normal behaviour from signal anomalies. Current challenges for DL approaches include the efficient composition of training datasets, particularly where operational conditions and environments vary, and training datasets cannot be representative of the full range, where uncertainty propagation is to be addressed by developing Monte Carlo simulation methods that have sufficient accuracy for non-linear, non-Gaussian, non-stationary stochastic processes. Furthermore, the data availability for training, variability of data sources, and accessibility or ownership rights challenge progress in developing DL processes for PHM systems.

For free-play SPHM, the use of artificial intelligence (AI) has merit in terms of machine learning and artificial neural networks, given the near non-deterministic nature of structural anomaly prognostics and gradual degradation rates [123]. Important to consider are the risks and regulatory challenges around “black-box” artificial intelligence approaches. Fairness, Accountability and Transparency (FAT) is an emerging requirement as AI becomes more widespread in use. This is covered in greater depth in the subsequent Section 4, which addresses, amongst other topics, operational decision-making [124]. A framework for real-time diagnostics and prognostics RUL of an electro-mechanical actuator (EMA) for aircraft flight control systems is proposed in [125], using on-board measurement of the motor current. This produced consistent and accurate results, including reconstructing on-board sensor signals using a combination of off-board physics-based knowledge, and on-board (on-wing) computationally efficient machine learning techniques for RUL estimation, such as Support Vector Machine (SVM) for rapid data assessment. The study found that the RUL estimate would underestimate with longer lead times, thus proposing future work is required for alternative machine learning approaches to capture the uncertainty propagation. However, it should be noted that next generation military aircraft rudders, flaperons and horizontal tails are powered by electro-hydrostatic actuator (EHA) systems [126]. Primary advantages of EHA systems are precise positioning with no error caused by gaps between mechanical components, meaning internally there are little to no backlash issues, and furthermore reduces the reliance on high maintenance, heavy central hydraulic systems. However, this does not exclude control surfaces from backlash or free-play faults, as there are several points of contact on the hinge line that can develop damage or fault, with high loads experienced on rudder hinges [127].

Taking a more general perspective, the stochastic nature of defect propagation and epistemic sources of uncertainty (see Section 4) generate significant prediction errors for model-based numerical simulations, where accurate long-time predictions are impossible to achieve without continuous adaptation and monitoring [128]. Although, where there are condition monitoring systems, predicting the degradation must factor sensor resolution and precision, as the variance needs to be small enough for a longer time horizon prognosis to be of use for maintenance scheduling. As supported in [21], understanding the failure mechanisms of the system that is being monitored is a major factor in estimating RUL, and a combination of model-based and data-driven methods shows promising results. The Digital Twin (DT) concept, a virtual digital version of a physical asset, makes use of parallel operations, enabling a DT to be run simultaneously to the mission or in advance of the physical model operating. This is explored in [129], where an aircraft engine that has 21 sensor inputs is continually updating RUL for the physical engine using a long short-term memory (LSTM) neural network. The dynamic model, which is based on the NASA C-MAPSS dataset, found that the LSTM approach yielded far better performance than similarity-based linear regression and an improvement on feed-forward neural network approaches. A subsequent study improved RUL estimation accuracy using C-MAPSS FD001 dataset employing an Adaptable Time Window–Acyclic Graph Network approach with Convolutional Neural Network (CNN)-LSTM [123]. A study of a non-deterministic prognostics approach using Monte Carlo methods, and high-fidelity finite element DT models, were used to generate four probabilistic estimates of crack state throughout the life of a geometrically complex test specimen [130]. The DT demonstrated accurate prediction of RUL to within 10% of the true test, while time to failure (or end of life) was within 0.2%. Incorporation of diagnostic data over time using Bayesian inference could reduce the variance in these predictions. Additionally, understanding the probability distribution function for component failure allows for scheduling maintenance, with optimal cost-benefit, and minimising the useful life waste of a component [131].

## 4. PHM-Enabled Decision Support for Military Fixed-Wing Aircraft Applications

Decision-making and decision support are accepted as an integral part of prognostics and health management. This is reflected in several PHM methodologies and process representations (e.g., OSA-CBM [132]) as well as review papers addressing the PHM state-of-the-art. All these sources serve to express that a crucial part of the PHM chain is to consider how any output from anomaly detection, diagnostics or prognostics algorithms is employed to deliver actionable information supporting meaningful, informed decision-making. The same considerations hold true for military aviation applications, with this section providing a more in-depth discussion of PHM decision support in this military context.

Various review papers provide a discussion of decision support in the context of PHM, with Bousdekis et al. [133] covering several decision-making requirements associated with CBM applications. It is highlighted that available input, desired output and user requirements related to accuracy, responsiveness and other aspects effectively combine to create a design space for decision methods and combinations of methods. Zio [57] provides a fairly comprehensive review of PHM model aspects with an influence on decision-making. This discussion covers the interpretability of models, in particular the robustness, causality and quantifiable reliability of outcomes and predictions to be used in decision-making. In addition, aleatory and epistemic sources of uncertainty are discussed, which include uncertainty regarding future operation profiles and state evolution. The latter aspects are mirrored by [134] and [42], who extend the discussion of uncertainty to include measurement uncertainty and modelling uncertainty.

These existing perspectives fall short in three important aspects: (1)Existing PHM literature—which covers both review papers and original research papers—does not account for the different time horizons, objectives and metrics involved in maintenance decision-making. As expressed by Bousdekis et al. [133], “the [decision] output can be either the optimal time for a pre-defined maintenance action or the optimal action and the optimal time for its implementation”. While this consideration is a good start, it foregoes a more in-depth discussion of the various time horizons and types of decisions associated with aircraft maintenance. In addition, the main objectives and associated metrics driving maintenance decision-making for military aviation applications have not been discussed in a comprehensive way in the state-of-the-art. Finally, while maintenance task determination and timing are the essential aspects of maintenance decision-making, there is little recognition of the constraints in terms of applicable regulations and standards.(2)Existing literature falls short in defining, standardising, and incorporating elements of uncertainty into the various stages of maintenance decision-making. As highlighted by Javed et al. [134] and Saxena et al. [31], it is “crucial to take into account the uncertainty in the prognostic output, especially when using them for decision making”. As such, the term ‘uncertainty’ and its characteristics should be defined in unambiguous terms. What kind of uncertainty is considered? Often sources of uncertainty are being confused or mixed, or it is not clear what kind of uncertainty is being characterised or quantified. Furthermore, with a lack of standardisation of methodologies in prognostics in general, standardisation of uncertainty—involving representation, quantification, propagation, and management [134]—is also far from achieved.(3)The existing literature generally does not account for a military aviation context, which has its own characteristics which set it apart from civil aviation applications.

In the following subsections, these challenges will be addressed in detail. 

### 4.1. Maintenance Decision-Making: A Multi-Level Perspective

To address the first challenge as highlighted above, it is first necessary to put PHM-enabled maintenance decision support in the context of applicable types of decisions and associated time horizons. Figure 6 represents the primary maintenance activities across a range of decisions, from strategic to tactical to operational. In addition, the associated time horizons are indicated using a scale from long-term to short-term. While the specific times associated with this scale will vary across different platforms with different lifespans, as well as different defence organisations, a typical representation is for strategic decisions to be coupled with time horizons in excess of 1 year (ranging up to the entire lifecycle of a platform, which may be 20+ years and sometimes far beyond that, as for instance for the B-52 Stratofortress). As such, strategic maintenance decision support involves considerations flowing into and out of platform acquisition, setting the stage for the platform capabilities and the associated strategic maintenance requirements. Examples include resource acquisition (including maintenance facilities, tooling, and know-how), as well as organisational preparedness. Strategic maintenance requirements flow down into fleet management decisions, where multi-year decisions regarding fleet objectives, composition, required maintenance resources and their strategic allocation are made [68,135,136,137]. 

Moving down in scope but closer to maintenance execution, tactical decisions typically involve time horizons between one month and a year, though it is not unusual for anything above one week to be considered in this category. Maintenance decision support for these types of decisions typically focuses on scheduling, in which assets and resources are allocated to generate the required mission readiness and flight hours for the considered unit of analysis (which may be at fleet, wing or squadron level). Managing a fleet, wing or squadron of aircraft is a dynamic and, at times, uncertain task, hence developing models to optimise fleet serviceability has numerous constraints [138]. More certainty on the parameters that are inputs to a model, for example component level data, can enhance the predictability of unscheduled maintenance events and, in turn, realise efficiency gains at an aggregated data level. Fleet management of military aircraft typically works around planning and scheduling of operations and maintenance, driven by aircraft availability, which differs from civil commercial aircraft more focussed on fuel burn and passenger and cargo profitability drivers [139,140]. While it is difficult to factor in discrete unscheduled maintenance events, the overall goal for military aircraft fleet managers is to ensure a steady-state flow or ‘stagger’ of aircraft flying hours in the fleet that moves into maintenance [141]. In the flight and maintenance planning phase, the allocation of tails to flights as well as maintenance is scheduled, helping to produce the required availability as well as ensuring stability in flight and maintenance operations, as studied in more detail in [138,142,143]. Aircraft fleet management simulation experiment modelling shows that fleet performance is most influenced by two factors, which are the management of unscheduled maintenance events, and the balance of flying loads across the fleet. Findings from simulation modelling carried out by [144] show that aircraft fleet management policies should first address the impact of unscheduled maintenance and tail rotation within the operation, as this will generate the greatest benefit to fleet performance over the serviceable life of the aircraft fleet. This is a significant driver for implementing SPHM technology, enabling greater lead times to address unscheduled maintenance before it impacts overall fleet management performance. From the SPHM perspective, the output of prognostics with sufficiently long prognostic horizons would consequently be most useful to be integrated into this process. Only a few studies are available which consider the relation of prognostics with scheduling in the military context [145,146,147]. 

Finally, operational decisions are associated with very short time horizons, typically looking at operations for the next week, with an emphasis on the next day or even decisions taken on the day of maintenance task execution itself. In relation to PHM, it is here that detection and diagnostic aspects are leveraged, with a number of studies (e.g., [148]) considering the output of anomaly detection and diagnostic algorithms and their potential role in maintenance decision-making. 

While the preceding discussion focused on the type of decisions and their operationalisation with respect to military (fixed-wing) aviation, in keeping with the third challenge identified above, it is necessary to understand the associated objectives, constraints and metrics which drive decision-making in this domain. To this end, Table 5 gives a comprehensive overview of these aspects drawing together information from various sources [138,142,150] as well as original insights. It is noteworthy that cost is not included into the objectives at any level; while cost is definitely a constraining factor in military maintenance, the minimisation of cost is usually not an explicit objective. Rather, military operators will try to increase their mission readiness (in its various forms) within a given budget.

Within the context of these types of decisions, PHM can influence all levels. At the strategic level, PHM has the potential to influence maintenance task requirements through substitution or escalation. In simple terms, the presence of a monitoring system may prevent the need to perform certain tasks, such as visual inspections or operational checks, with the PHM capability taking over these responsibilities. If tasks cannot be substituted, the PHM capability may generate sufficient trust in the estimation and prediction of health to allow for the escalation of maintenance intervals, extending the periods between maintenance tasks such that the overall maintenance requirements decrease. On the other hand, including a PHM capability into a fleet of aircraft generates its own challenges, usually associated with additional power requirements and weight implications, as well as the need to develop, operate and maintain the monitoring system. Task substitution or escalation percolates down to the tactical level, where the scheduling effort benefits from decreased task requirements. Several original research papers consider the implications of task requirements and long-term uncertainty at the fleet management level for military applications, such as the work by Mattila et al. [151] on various techniques to analyse fleet management issues in the Finnish Air Force. Particular challenges at the tactical level include how to deal with uncertain or incorrect predictions, a topic which will be covered in more detail in Section 4.2. From the military domain perspective, Mikat et al. [152] have performed an evaluation of different options for dynamic mission and maintenance scheduling. Marlow et al. [153] study the application of various optimisation techniques to help address problems with unscheduled maintenance effects in military aircraft fleet management, applying these techniques to “both day-to-day planning and medium-term forecasting”. As such, their work crosses over into the operational level. At this level, PHM capabilities may support the detection and troubleshooting of degradation, defects and/or failure events. This is evident by the text clustering analysis highlighted in Section 2.2, showing the two major clusters D and E include relevant terms for PHM decision-making and degradation in faults among the top ten most frequent terms. Possible benefits include the detection of hitherto unknown defects or events, as well as reduced time to figure out what is going wrong with an aircraft, or, more typically, its components, subsystems and/or systems. Furthermore, PHM may increase decision capability through advanced visualisation methods, integrating various decision inputs with multi-criteria decision-making methods to offer decision alternatives for the operator or technician to select [154] and reducing the time for required maintenance interventions [81,155,156,157,158,159]. Of particular relevance are recent advances in mixed reality, which is an emerging technology and can support better comprehension of complex tasks and enhance in situ decision-making for maintenance practitioners [160,161,162]. Currently, mixed reality has had limited implementation into aircraft sustainment, given the technology is an emerging tool at a TRL of 6 [163], balancing actual need, fit-for-purpose, “fashnology” and operational risk factors [164]. However, for training purposes there is early adoption, with aircraft technicians utilising Microsoft HoloLens head-mounted devices for remote communication with technical experts, for example, the Royal Australian Air Force (RAAF), together with overseas Boeing technicians used the HoloLens II primarily as a remote assistance tool for Boeing C-17A Globemaster III maintenance tasks [165], proving to be a collaborative tool for training and familiarisation [166]. 

### 4.2. Uncertainty in PHM-Enabled Maintenance Decision-Making

To use PHM output in maintenance decision-making, it is crucial to incorporate the fact that model inputs and outputs will have a certain degree of uncertainty. Unfortunately, existing literature falls short in defining, standardising, and incorporating elements of uncertainty into the various stages of maintenance decision-making, especially when considered within the military domain.

In terms of definition, uncertainty is defined and interpreted across a wide range of domains and applications, as set out comprehensively by Thunissen [167]. Consequently, there are substantial differences to be considered, and one cannot assume that the mention of uncertainty in the context of research leads to an unambiguous understanding. As highlighted by Thunissen [167], within decision-making research, uncertainty is typically simplified into representing “risk and uncertainty (ignorance)”. Risk describes a situation where each action may lead to one outcome from a set of possible outcomes, with each outcome occurring with a known probability. Therefore, all possible actions are known, all possible outcomes arising from an act are known, and the associated probabilities can be quantified. Uncertainty represents a situation where either action or a combination of action and outcome has a set of possible outcomes which cannot be quantified probabilistically. Notably, this definition is quite different from that which is prevalent in engineering, which is the category of research with which PHM is most closely aligned. Here, uncertainty represents incompleteness of knowledge which drives a difference between model-based predictions and reality, or as defined by [168]: “the incompleteness in knowledge (either in information or context), that causes model-based predictions to differ from reality in a manner described by some distribution function”. As a third option, uncertainty is characterised into different categories, each having its own definition [167]. These different types are ambiguity, epistemic uncertainty, aleatory uncertainty, and interaction uncertainty. In particular, epistemic and aleatory uncertainty are frequently highlighted in discussions of PHM uncertainty and associated uncertainty quantification, where others propose to recognise ontological uncertainty as well [169]. Epistemic uncertainty represents any lack of knowledge or information in any phase or activity of the modelling process, where the fundamental cause is incomplete information or incomplete knowledge of some characteristic of the system or the environment. Further subdivisions in epistemic uncertainty incorporate model, phenomenological and behavioural uncertainty. Aleatory uncertainty represents “inherent variation associated with a physical system or environment under consideration” [167], which cannot be controlled by a decision-maker. Ontological uncertainty is uncertainty due to completely unknown factors [169], which is relevant for PHM systems developed for military applications which may have lifespans ranging into decades; as Dewey et al. [169] note, “unknown or poorly categorized phenomena [may] dominate the response of the system in the far term”.

Within PHM-enabled decision-making, several sources of uncertainty are acknowledged. Javed et al. [134] identify “input uncertainty from system, measurement uncertainty from sensors, operational environment uncertainty from usage conditions, and modelling uncertainty from degradation model”. They note that any of these sources of uncertainty will impact the accuracy of RUL predictions and the associated decisions, highlighting that a certain level of confidence is required to enable offline or online decisions. Similar sources of uncertainty are identified by Fink et al. [42], who highlight that degradation behaviour may not be exactly known (with models that are an approximation), measurement errors may occur, and future operating profiles and loading may be unknown. Another alternative is posited by Dewey et al. [169], who highlight model input uncertainty, model discretisation uncertainty, and model form uncertainty as the main sources of uncertainty. In this perspective, model input uncertainty is the uncertainty in any input to the mathematical models used to perform the analysis, whereas model discretisation uncertainty is “the uncertainty in the implemented mathematical model due to the finite resources available on computer systems” [169]. Finally, model form uncertainty relates to the uncertainty in the degree to which the implemented mathematical model represents the real-world behaviour of the physical objects that are represented in the model (via assumptions). 

To tackle the issue of uncertainty, Javed et al. [134] propose the following steps:**Represent uncertainty**: the representation of uncertainty involves the choice of modelling and/or simulation approach. Within the PHM domain, a probabilistic representation of uncertainty is most commonly adopted. As Fink et al. [42] note, estimates should “at the very least be accompanied by confidence intervals and, which is even better, by a description through probability distributions if at all possible, or by fuzzy representations.”.**Quantify uncertainty:** Dewey et al. [169] define uncertainty quantification as ”the combination of verification (assessment of mathematical accuracy) and validation (assessment of applicability) of mathematical models of real-world phenomena”. They highlight that uncertainty quantification is a requirement for PHM as the purpose of a PHM system is to ascertain the reliability of an asset via probabilistic methods, and furthermore assert that “those working in the field of PHM have traditionally quantified sources of uncertainty from the aleatory risk of a component in their analyses while completely ignoring other sources of uncertainty that may occur from epistemic risks” [169]. Javed et al. [134] note that the quantification of uncertainty involves the identification and inclusion of different sources of uncertainty in the most accurate and reliable way possible.**Propagate uncertainty:** importantly, uncertainty is not (just) a point measure, it acts and potentially grows over time. Certainly, with PHM applications, it is natural to expect predictions to include additional variance the longer the prognostic horizon will be, as noted by Mikat et al. [152]. Uncertainty propagation accounts for a time-based representation of previously quantified uncertainties, which is used to predict future states and their uncertainty, as well as estimate RUL and its uncertainty. These considerations are particularly relevant for the tactical maintenance decision support phase, where the scheduling of maintenance activities is highly dependent on the level of uncertainty over time, as well as the tolerance of the maintenance system to accommodate for this uncertainty.**Manage uncertainty:** the representation, quantification, and propagation of uncertainty open up the possibility of proactively managing the uncertainty of future states and RUL estimates. As noted by Javed et al. [134], the quality, reliability and configuration of sensors may help to decrease the uncertainty, as well as improve modelling for health assessment and prognostics, for instance, through hybrid approaches, which decrease epistemic uncertainty regarding underlying physical behaviour.

In the military domain, very limited examples exist of research works that span the bridge from PHM to decision support while incorporating uncertainty. Early work by Byington et al. [170] considered some novel applications of proposed prognostic enhancements to diagnostics systems. In more recent work, Vandawaker et al. [171] have examined the impact of prediction accuracy uncertainty in remaining useful life prognostics for a squadron of 12 aircraft, with an uncertainty factor being applied to the useful life prediction and subsequently explored through discrete event simulation covering pre-flight, flight, and post-flight operations, as well as maintenance and logistics activities. Results are compared to a baseline case constituting traditional time-driven maintenance. Somewhat similarly, Macheret et al. [172] analyse and compare overhaul and prognostic asset management strategies for military platforms. The authors highlight that prognostics has the potential for improved operational availability at a significantly lower cost (as expressed via a number of spares) compared to that of the overhaul maintenance strategy, as well as reduced risk of failure due to informed decision-making on asset selection for upcoming missions.

## 5. Future Challenges and Opportunities for Predictive Maintenance in Military Aviation

Despite recent advances in predictive maintenance driven by new aircraft designs, increased sensor capabilities and digitalisation of operations and maintenance processes, and improved models and algorithms, there are some major challenges to address before predictive maintenance, SPHM and similar initiatives will have a major impact within defence sustainment and operations. 

In line with the order of discussion in this review, the following challenges can be distinguished:Several challenges focus on the availability and suitability of data, which relate to sensor capabilities on aircraft as well as the supporting data (pre-)processing infrastructure and processes.The availability and (long-term) reliability of sensors on military aircraft has to be ascertained per platform. Not every aircraft type has the same capabilities in terms of data capture and storage. Legacy platforms typically have less—or less precise—sensors, which are typically geared towards aircraft control purposes rather than being purposely designed to support predictive maintenance.The specific topology of sensors or sensor networks on specific platforms may preclude the generalisation of models towards other platforms (e.g., what works on the F-35 Joint Strike Fighter may not work on the F-22 Raptor).There are several challenges related to data integration: acquiring data is not as straightforward as it sounds on paper, whether that is due to complexities in data acquisition systems or a lack of necessary infrastructure to record and transmit data to maintenance engineers.Processing aircraft data is still a challenge for operators, even if big data analytics are feasible, a point which is related to having sufficient skilled labour to realise the full potential within the gathered data.

As highlighted in a broader context by Zio [57], the next step in the SPHM chain is fault/damage detection, the success of which is strongly impacted by the quality of the features selected by pre-processing. As Zio [57] notes, “unfortunately there is no universal rule for choosing the optimal pre-processing method”, an observation which is as true for military aviation applications as it is for other domains. A further complication is the fact that various epistemic uncertainties start to apply in the detection stage to make signal processing a major challenge. In the defence context, several examples of this have been highlighted in the discussion of non-linearities for free-play detection.

For diagnostics, the application of SPHM is also impacted by the presence of uncertainty which derives from the processing of data from measured sensor signals. The additional variance introduced by the more challenging load spectra of military aircraft makes this issue even more pressing. Within the military context, individual aircraft tracking (including loads tracking, environmental exposure tracking, etc.) is strongly advisable to counter this issue of variance. Tracking and subsequent condition monitoring of engine component systems are well established and applied; however, tracking of structural components and subsystems is typically not yet implemented for individual aircraft. Another issue is formed by long-term dependencies in tracking data, which are difficult to account for in most diagnostic algorithms. Yet, with the long lifespans of military platforms, which include major life events, such as mid-life upgrades, these dependencies have a high probability of being present. Beyond these considerations, successful diagnostics typically relies on the availability of high-quality labelled data regarding failure or damage events. To detect, localise and characterise what happened (for instance, in terms of failure mode determination), data labelling is required, but this is typically difficult, expensive and labour-intensive [57]. 

In terms of prognostics, a primary issue relates to the future usage and mission profiles of military aircraft, as well as epistemic and ontological uncertainty regarding the operational environment in which these aircraft will operate. Some theatres of operation can be predicted well during the design phase of military aircraft, but other theatres may be outside of the original design spectrum. 

Another issue is the effect of missing data on prognostics performance. As highlighted previously, individual flight tracking is not always established yet, let alone the supporting infrastructure to consistently capture condition data over the life of an aircraft (which may, for instance, require data capture at remote outstations in difficult theatres of operation). What is more, very few research works tackle the issue of missing data and their effect on prognostics performance. For the military context, while a comprehensive review of prognostics [52] identified there are few case studies published that apply prognostics to real-world problems in realistic operational environments, the fact that current industry-based prognostics widely use trend extrapolation poses a concern as it is often the least accurate method and not considered sufficient for practical purposes.

The latter point impinges on some of the challenges related to decision-making using PHM solutions. First of all, there is a lack of standards for SPHM technology implementation for military fixed-wing aircraft. While some initiatives are underway for civil aviation applications (e.g., IP-180 and ARP 6461) [61,173], similar developments have not yet reached the mainstream for the military domain at a direct implementation level, despite the fact that the US DoD has issued a CBM+ guidebook [80,116]. Guidelines standards [132] for SHM and predictive maintenance technologies specifically relevant to unscheduled faults have emerged in recent years for civil aviation; however, military standards do not exist, but are rather focussed more on longer time horizon faults, such as airframe fatigue life.

The opportunity for research originality and new contributions to the body of knowledge is high in the discussed areas of aircraft SPHM. The following section discusses the rationale for this research with current industry relevance. Current reliability practices are, in some ways, inherently reactive [64], limited by the reliance on analysis of system faults for discrete occurrences, a measure based on prior maintenance events. As demonstrated by [70], 89% of system failures cannot benefit from simple rules that limit the system operating age; as such, PHM technologies can offer lead indicators to reliability measures and perhaps a change to reactive maintenance practices.

Defence operators at a strategic level are preparing for, and, in some cases, ordering [79] predictive maintenance practices; however, the PHM architecture at the operational level is still in the process of maturing. Firstly, the acquisition of on-board sensor data at a system level requires greater integration with central maintenance management systems and to become readily accessible to engineers performing the analysis [77]. Secondly, the airworthiness program structures at a tactical level need to facilitate diagnostic capabilities and AI/machine learning approaches in areas of prognostics: (i) run to failure, (ii) comparative failure data, (iii) known threshold of failure. Furthermore, at a tactical level the trust in data becomes critical, with a preference for quality data over quantity, which is stressed by [120], “when failure is unpredictable due to randomly changing conditions, then RUL becomes meaningless and maintenance decisions are based on current condition”. It remains a challenge that the cost of maintaining sensor systems on-board military aircraft has traditionally precluded them from being deployed across individual aircraft or fleet wide. Another issue is the required maintenance to ensure the sensors are calibrated and functioning. Challenges in the transition from initial detection and diagnostics to prognostics are underpinned by the quality of RUL: “The management of uncertainty is an important and often overlooked aspect in the estimation of [RUL].”. As noted by Engel et al. [119] and going on to explain “precise estimates of RUL have a very low probability of being correct”. “In fact, the remaining life estimate with the widest confidence interval (lowest precision) may offer the least unnecessary maintenance”. Related to this, there are several opportunities to progress decision-making on the basis of PHM output in military aviation applications. First of all, the aforementioned transition challenges towards prognostics and RUL estimation may be influenced by the development of more capable, yet less computationally intensive, machine learning approaches for prognostics and overall health assessment, for fleet management decision-making. This does, however, hinge on the gap between decision-making in practice (whether at the strategic, tactical, or operational level) and the maturity of PHM solutions. A related aspect is that currently used accuracy and performance metrics [31] in predictive maintenance / PHM do not necessarily translate well to decision-making criteria and metrics in military aviation applications. As highlighted in Section 4, different objectives, constraints, and metrics apply, and a fruitful direction for future research will consider bridging this gap. When prognostics do reach the work floor, visualisation of SPHM information at the engineering department level as well as in situ can influence the effectiveness of decision-making as identified in [46]. However, it will be of paramount importance to investigate and develop ways to secure the interpretability, security and trust associated with predictive maintenance solutions, so that acceptance of prognostic tools becomes a well-founded aim rather than lucky happenstance. 

The ability to quantify uncertainty across tactical and strategic timescales is limited due to the uncertainty of prognostics with long-time horizons, in particular for military aircraft [145,146,147]. In such cases, changes in sensor capabilities and topology, individual aircraft tracking variabilities, changing mission profiles and locations, and mid-life upgrades can all impact decision-making uncertainty for lifespans ranging into decades, as noted by Dewey et al. [169]. Consequently, over such long-time horizons, changes in decision-makers brings into question the varying interpretability of prognostics, security environments, trust and acceptance of prognostic tools; this may introduce a requirement for greater transdisciplinary approaches beyond engineering, involving behavioural science or organisational culture. Moving to a strategic level, PHM may influence maintenance tasks through substitution or escalation. As previously discussed, the challenges of not having a rigid standard practice for emerging PHM capability may introduce uncertainty in the control and accountability of maintenance tasks. Decision-making at each stage of the predictive maintenance process will have uncertainty to varying degrees from the sensor system, usage environment, measurement errors and modelling of degradation, which is noted by Javed et al. [134] and Fink et al. al. [42]. This places emphasis on the decision-making process to factor in levels of confidence, and it should be known and monitored to constrain uncertainty in the overall PHM tool.

In summary, these three areas discussed here aim to address and progress prognostics for defence aircraft sustainment and operations, highlight the research gaps that leave limitations in advancing SPHM from higher technology readiness and implementation. However, readers should be confident in the opportunities and potential outcomes, with analysis by Macheret et al. [172] showing prognostics for military aircraft can improve operational availability, result in lower maintenance costs, and reduced risk of failure due to informed decision-making.

## 6. Conclusions

Predictive maintenance has become an area of significant interest for researchers and practitioners. While significant advances have been made, it is abundantly clear from a variety of recent review papers that much work remains to be conducted. This impression is reinforced by the systematic review performed in this work, which addresses the lack of any review to date relative to predictive maintenance for military fixed-wing aircraft in the defence context. Through analysis of fifty recent review papers, along with a macro-view of publication trends in the overall state-of-the-art and discussion of specific application papers, areas of interest have been delineated and discussed in detail in this review. Challenges, opportunities, and future research directions are discussed, collating the state-of-the-art and identifying trends that demonstrate the challenges that predictive maintenance faces in the short-term and long-term, as decisions are made sometimes 20 years in advance over platform capabilities, as in the case of defence.

Predictive maintenance in defence settings is forging a path forward to higher TRLs and implementation, as the aircraft platforms provide opportunities for prognostics to be performed. Predictions generated from prognostics are, in many ways, just a starting point, and subsequent implementation at scale requires embedding into various levels of decision-making, with operational use in particular being susceptible to issues with trust and acceptance of proposed solutions. This could be compounded with the future of autonomous aircraft, where removing pilots removes a source of data and inevitably sustainment and operations will have to trust and rely on the aircraft sensory systems. A transdisciplinary and whole of systems approach is required to integrate predictive maintenance into the full lifecycle of maintenance practices and ensure it is effective overall in defence sustainment and operations. Nevertheless, there is an opportunity in the near-term for defence fixed-wing aircraft predictive maintenance, and the recent literature demonstrates it is tackling the higher TRL challenges. Lastly, military and civil domains may see benefit from greater cross-collaboration, enabling a dual-use case for predictive maintenance technologies, while also providing higher returns on funding and resources.

## Figures and Tables

**Figure 1 sensors-22-07070-f001:**
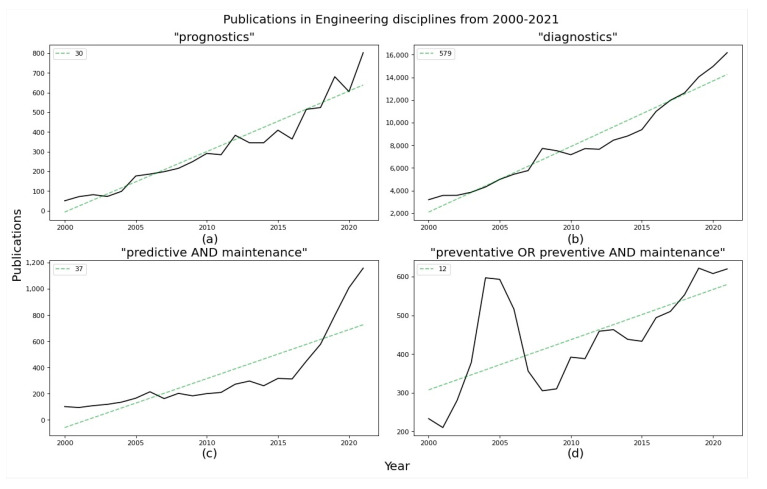
Publications in engineering disciplines from 2000–2021 that include terms: (**a**) “Prognostics”; (**b**) “Diagnostics”; (**c**) “Predictive Maintenance”; (**d**) “Preventative OR Preventive AND Maintenance”. Includes slope (dashed line), a measure of the rate of increase in publications over time (source: Scopus).

**Figure 2 sensors-22-07070-f002:**
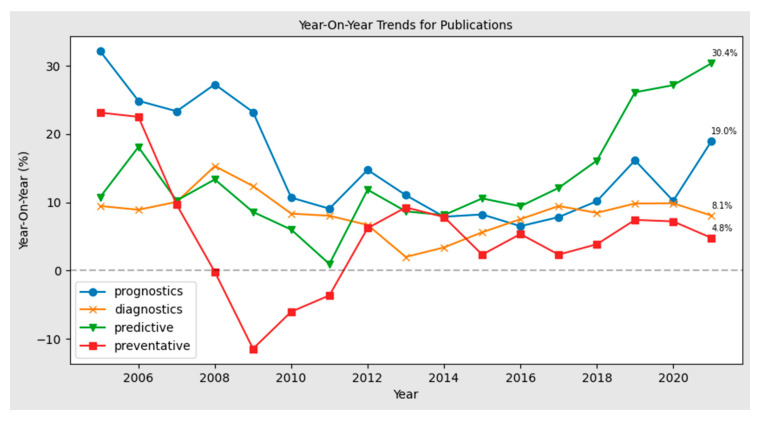
Year-on-year trend comparison of publications at a rolling average of five years for the period 2000–2021: “Prognostics”; “Diagnostics”; “Predictive Maintenance”; “Preventative OR Preventive AND Maintenance” (source: Scopus).

**Figure 3 sensors-22-07070-f003:**
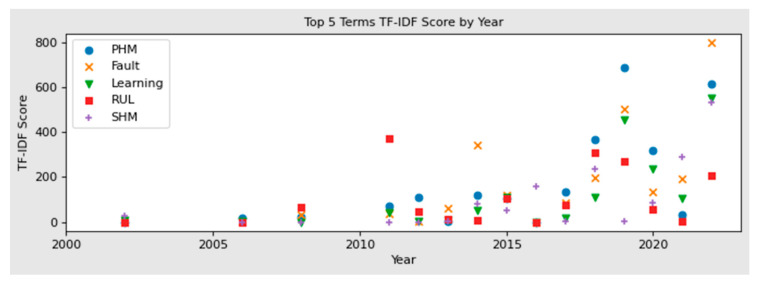
TF-IDF scores for the top five terms across the fifty review papers grouped by year.

**Figure 4 sensors-22-07070-f004:**
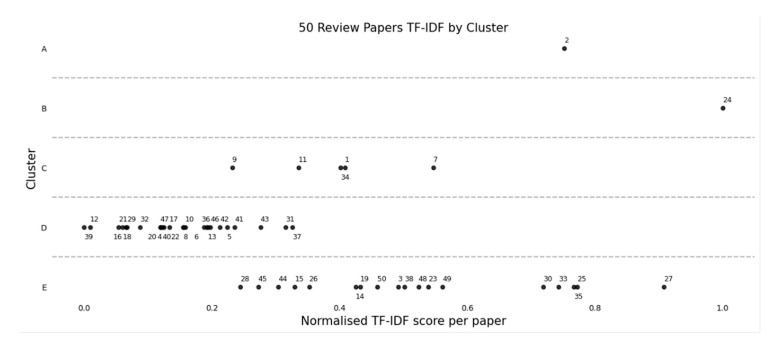
Fifty review papers analysed by TF-IDF and clustered using K-means, subsequently plotted within respective clusters against a normalised TF-IDF score per paper.

**Figure 5 sensors-22-07070-f005:**
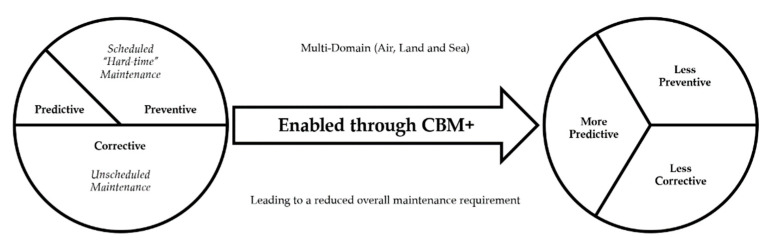
Representative illustration of the United States Department of Defense Maintenance Strategy Transition—“Enabled through CBM+” [80].

**Figure 6 sensors-22-07070-f006:**
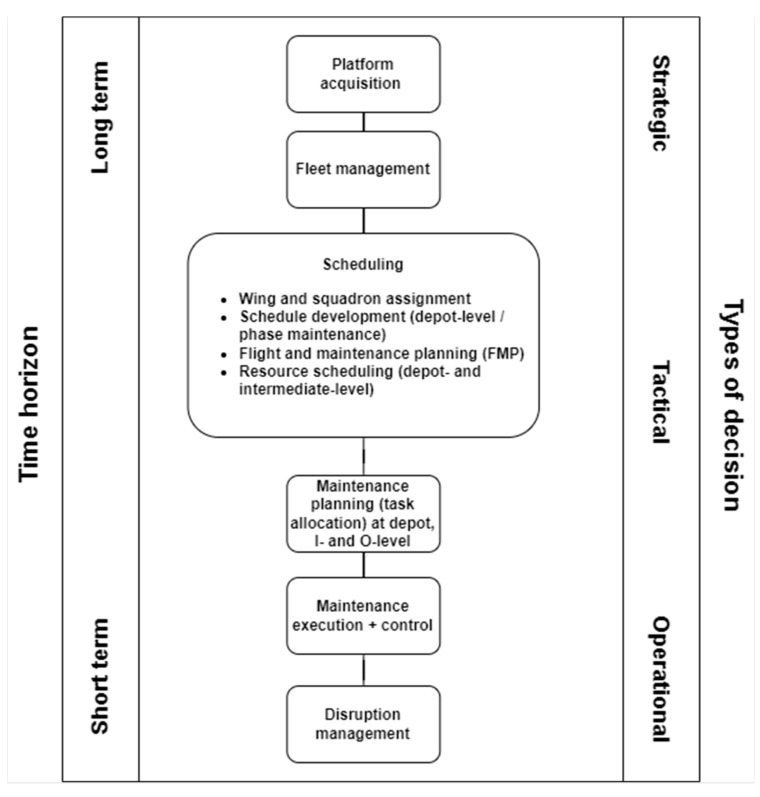
Defence maintenance activities across time horizons and types of decision (adapted from [149]).

**Table 1 sensors-22-07070-t001:** Fifty recent review publications analysed using TF-IDF with terms ranked in order by the highest total score for all papers across the entire corpus.

Rank	Term	Rank	Term	Rank	Term	Rank	Term
1	PHM	6	Damage	11	Composite	16	Decision
2	Fault	7	Structural	12	Diagnostics	17	Driven
3	Learning	8	Prognostics	13	Machine	18	Nonlinear
4	RUL	9	Engine	14	Noise	19	Turbine
5	SHM	10	Degradation	15	Deep	20	Reasoning

**Table 2 sensors-22-07070-t002:** Analysis key parameters used for TF-IDF and K-means text clustering.

TF-IDF Key Parameters:	K-Means Key Parameters:
ngram_range = (1, 1) stop_words = English + [(e.g., “http”, “doi”, etc.)] max_features = 100 max_df = 0.8 min_df = 5	n_clusters = 5 random_state = 0 max_iter = 10,000 init = “k-means++” n_init = 10

**Table 3 sensors-22-07070-t003:** Top ten terms of fifty review papers using TF-IDF and K-means clustering, with reference to corresponding numbers from Figure 4 and supporting references to papers in brackets [].

Cluster A	Cluster B	Cluster C	Cluster D	Cluster E
Noise Engine Fault Power Turbine Gas Vibration Diagnostics Engines Frequency	Gas Turbine Engine Fault Diagnostic Vector Linear Parameter Fuzzy Simulation	SHM Composite Damage Wave Structural Structures Fusion Figure Inspection Frequency	RUL Decision CBM Structural PHM Prognostic SHM Fault Fuel Degradation	PHM Fault Learning RUL Deep Diagnosis Prognostic Reasoning Machine Degradation
2[9]	24[10]	1[11], 7[1], 9[12], 11[13], 34[14]	4[15], 5[16], 6[17], 8[18], 10[19], 12[20], 13[21], 16[22], 17[23], 18[24], 20[25], 21[26], 22[27], 29[28], 31[29], 32[30], 36[31], 37[32], 39[33], 40[34], 41[35], 42[36], 43[37], 46[38], 47[39]	3[40], 14[41], 15[42], 19[43], 23[44], 25[45], 26[46], 27[47], 28[48], 30[49], 33[50], 35[51], 38[52], 44[53], 45[54], 48[55], 49[56], 50[57]

**Table 4 sensors-22-07070-t004:** Review papers, articles, and standards relevant to predictive maintenance.

Type	Year	Relevance	Ref.
Review	2022	Directions for assisting researchers and practitioners in advancing PHM methodologies and maturing practical PHM technologies.	[43]
Review	2022	Extensive review of key advancements and contributions to knowledge in the field of Integrated System Health Management for the aerospace industry, with a particular focus on various architectures and reasoning strategies involving the use of artificial intelligence.	[50]
Review	2022	Aims at pointing out the main challenges and directions of advancements in PHM, for full deployment of condition-based and predictive maintenance in practice.	[57]
Review	2019	Reviews the challenges, needs, methods, and best practices for PHM within manufacturing systems.	[53]
Review	2015	Shed light on the various maintenance models and their use in real-world applications, exploring the gap between academic research and practice.	[58]
Article	2018	Discusses the evolution of maintenance, the goals of the various stakeholders and implementation of PHM at commercial airlines.	[54]
Article	2013	Summary of SAE ARP 6461A [62] guidelines focuses on the key steps needed to implement SHM technologies within the regulatory environment and prevailing aircraft structural design and maintenance practices.	[61]
Article	2006	ISO Standards for Condition Monitoring, outlining processes for condition monitoring system design and implementation of diagnostics and prognostics.	[60]
Article	2005	Technical overview of Integrated System Health Engineering and Management (ISHEM) outlines a functional framework and architecture for ISHEM operations, describes the processes needed to implement ISHEM in the system lifecycle, and provides a theoretical framework to understand the relationship between the several aspects of the discipline.	[59]
Standards	2021	SHM standards applicable to civil aerospace, for stakeholders seeking guidance on the definition, development, and certification of SHM technologies for aircraft health management applications.	[62]

**Table 5 sensors-22-07070-t005:** Maintenance decision-making objectives, constraints, and metrics for military aviation applications.

Decision Types	Objectives	Constraints	Metrics
Strategic	Deployment capability Responsiveness Mission readiness Logistic footprint Lifecycle cost	Budget Workforce composition, size and training Base facilities and positioning Spare parts supply chain (parts availability/obsolescence) Mid-life upgrades/major modification programs	Operational expenditure Maintenance expenditure Fleet status
Tactical	Fleet availability Fleet health/reliabilityAircraft availability	Budget Space/facility constraints Workforce availability, skills Inventory status Spare part lead times Usage profiles	Availability Serviceability Sustainability Flight hour requirements + production
Operational	Aircraft availability Minimise unnecessary maintenance	Spare parts/component availability Person power availability (incl. skills) Minimum number of daily spare aircraft Minimum and maximum daily flight hours	Serviceability (instantaneous availability) Residual Flight Time (RFT)

## Data Availability

No new data were created or analysed in this study. Data sharing is not applicable to this article.

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
