# Peer review of "A Systematic Literature Review of Predictive Maintenance for Defence Fixed-Wing Aircraft Sustainment and Operations"

_sensors, 2022, doi:10.3390/s22187070_

Round 1

Reviewer 1 Report

The work is a complete overview of the current state and issues related to predictive maintenance enabled by PHM systems for Defence Fixed-Wing Aircraft. Specifically, it provides an interesting approach to analysing papers using machine learning and natural language processing algorithms.  

Some considerations for improvements:

1. Paragraph 2.2: the authors state " These papers have been selected for relevance...from several hundred papers to the selected fifty". I think that this statement should be demonstrated by showing to the reader the process of selecting papers, e.g through the systematic literature review method. 

2. Paragraph 2.2: seems there is not a complete description of the parameters cited in Table 2. It is recommended to insert such a description to facilitate the comprehension of the method to the reader. Moreover, although the content shown in Figure 4 is clear, it is not enough clear how the authors normalize the TF-IDF score per paper. It is suggested to clarify this step. 

3. It is recommended to insert in the sections regarding the Structural Health Monitoring in military fixed-wing aircraft and the Diagnostics approaches for military fixed-wing aircraft applications the following works:

Perfetto D., Sharif-Khodaei Z., De Luca A., Aliabadi M.H., Caputo F., Experiments and modelling of ultrasonic waves in composite plates under varying temperature. Ultrasonics, Vol. 126, 2022, 106820

Perfetto D., De Luca A., Perfetto M., Lamanna G., Caputo F., Damage Detection in Flat Panels by Guided Waves Based Artificial Neural Network Trained through Finite Element Method. Materials, Vol. 14(24), 2021, 7602 (Topic: ANN for SHM)

De Luca A., Perfetto D., Lamanna G., Aversano A., Caputo F., Numerical Investigation on Guided Waves Dispersion and Scattering Phenomena in Stiffened Panels. Materials, Vol. 15(1), 2022, 74 (Topic: FE models for SHM)

De Luca A., Perfetto D., Polverino A., Aversano A., Caputo F., Finite Element Modeling Approaches, Experimentally Assessed, for the Simulation of Guided Wave Propagation in Composites. Sustainability, Vol. 14(11), 2022, 6924

De Luca A., Perfetto D., De Fenza A., Petrone G., Caputo F., A sensitivity analysis on the damage detection capability of a Lamb wavessed SHM system for a composite winglet. Procedia Structural Integrity, Vol. 12, 2018, pp. 578-588

Author Response

Our thanks to the anonymous reviewers for providing their constructive suggestions. Please find the review comments below in regular font, and our response in italics. .

Reviewer 1

The work is a complete overview of the current state and issues related to predictive maintenance enabled by PHM systems for Defence Fixed-Wing Aircraft. Specifically, it provides an interesting approach to analysing papers using machine learning and natural language processing algorithms.  

Thank you for your comment.

Some considerations for improvements:

  1. Paragraph 2.2: the authors state " These papers have been selected for relevance...from several hundred papers to the selected fifty". I think that this statement should be demonstrated by showing to the reader the process of selecting papers, e.g through the systematic literature review method. 

We have updated the relevant text in Paragraph 2.2 to give additional insight into the selection criteria and process to arrive at the mentioned 50 papers. The updated text is given below and can furthermore be found in the tracked changes in the revised manuscript.

“These papers have been manually selected for relevance to the domain of aircraft predictive maintenance through an initial selection by keyword, subsequent abstract review, followed by full paper review, bringing down the sample from several hundred papers to the selected fifty. This down-selection was performed systematically through identification of papers published within the past twenty years, focusing on review papers only. Furthermore, these review papers were only selected if they had relevance to aircraft maintenance or Defence relevance in the context of fixed-wing aircraft, in line with the scope of this review. The relevance was established by a thorough review of individual review papers, including their stated scope, aims and objectives, and application domain(s).”

  1. Paragraph 2.2: seems there is not a complete description of the parameters cited in Table 2. It is recommended to insert such a description to facilitate the comprehension of the method to the reader. Moreover, although the content shown in Figure 4 is clear, it is not enough clear how the authors normalize the TF-IDF score per paper. It is suggested to clarify this step.

Thank you for these helpful comments. We have inserted a description of the parameters in Table 2 as suggested. Again, please refer to the text below or the tracked changes in the revised submission. Furthermore, relative to the normalisation of TF-IDF scores, a clarification is added through the inclusion of the applied equation (labelled equation (4) in the revised submission) – see tracked change on page 7, just above Figure 4. 

“The TF-IDF analysis key parameters include limiting terms to only single whole words, excluding terms made up of two or more words. To clarify this, only unigrams have been considered, where the term contains only a single word in sequence. For example, the TF-IDF analysis returned the unigram terms “PHM” and “prognostics”, in contrast to the potential for n-grams such as “prognostics health management”.  Furthermore, a library of common English stop words is used to exclude terms that are insignificant such as articles, pronouns, prepositions, conjunctions, and additional terms that are artefacts of the document processing, such as “http” and “doi”. The TF-IDF analysis is constrained to 100 terms, so sufficient comparison within the corpus can be made, and manual fine-tuning of parameters is made easier with a review of these mid to high-ranking terms. The key parameters “max_df” and “min_df” ignore corpus terms with maximum and minimum thresholds respectively which subsequently decreases processing time. Parameter “max_df” is set to 0.8, which ignores terms that appear in more than 80% of the documents, in turn removing terms that appear too frequently specific to the corpus. Similarly, parameter “min_df” is set to 5, which ignores terms that appear in less than five documents, which removes terms that appear too infrequently.

Using the K-means clustering machine learning algorithm [8], five clusters are set, with key parameter “n_clusters”, a simple nominal amount for ease of analysis, and for reproducibility, a “random_state” parameter of zero was selected to ensure deterministic results. It should be noted that the randomisation parameter was checked for multiple distinct random seeds, and the results were found to be stable. Furthermore, as shown in Table 2, the key parameter “max_iter” was set to 10,000, to ensure consistency across runs. While “n_init” is default at ten runs for varying centroid seeds and “init” set to “K-means++” also a default parameter with faster convergence, compared with a random initialisation of centroids.”

  1. It is recommended to insert in the sections regarding the Structural Health Monitoring in military fixed-wing aircraft and the Diagnostics approaches for military fixed-wing aircraft applications the following works:

Perfetto D., Sharif-Khodaei Z., De Luca A., Aliabadi M.H., Caputo F., Experiments and modelling of ultrasonic waves in composite plates under varying temperature. Ultrasonics, Vol. 126, 2022, 106820

Perfetto D., De Luca A., Perfetto M., Lamanna G., Caputo F., Damage Detection in Flat Panels by Guided Waves Based Artificial Neural Network Trained through Finite Element Method. Materials, Vol. 14(24), 2021, 7602 (Topic: ANN for SHM)

De Luca A., Perfetto D., Lamanna G., Aversano A., Caputo F., Numerical Investigation on Guided Waves Dispersion and Scattering Phenomena in Stiffened Panels. Materials, Vol. 15(1), 2022, 74 (Topic: FE models for SHM)

De Luca A., Perfetto D., Polverino A., Aversano A., Caputo F., Finite Element Modeling Approaches, Experimentally Assessed, for the Simulation of Guided Wave Propagation in Composites. Sustainability, Vol. 14(11), 2022, 6924

De Luca A., Perfetto D., De Fenza A., Petrone G., Caputo F., A sensitivity analysis on the damage detection capability of a Lamb wavessed SHM system for a composite winglet. Procedia Structural Integrity, Vol. 12, 2018, pp. 578-588

Thank you for these suggestions. We have carefully reviewed these papers and have included the following two references (labelled 86 and 87) into the narrative, in line with specific applicability for the review scope and objectives:

Perfetto, D.; De Luca, A.; Perfetto, M.; Lamanna, G.; Caputo, F. Damage Detection in Flat Panels by Guided Waves Based Artificial Neural Network Trained through Finite Element Method. Mater. Basel Switz. 2021, 14, 7602, doi:10.3390/ma14247602.

De Luca, A.; Perfetto, D.; Polverino, A.; Aversano, A.; Caputo, F. Finite Element Modeling Approaches, Experimentally Assessed, for the Simulation of Guided Wave Propagation in Composites. Sustainability 2022, 14, 6924, doi:10.3390/su14116924.

Reviewer 2 Report

According to my opinion this paper requires extensive revisions. Abstract and conclusions does not seem of standard profession level. Authors are advised to rewrite the abstract and conclusions sections. Also this paper contains a lot of unnecessary basic information which researchers already know, so focus on the key points more rather than providing the introductions.

It would be better if you use more Figs. to explain your work. It is not necessary to copy paste the diagrams from others, you can use your own block level diagrams after going through the related research material.  

Author Response

Our thanks to the anonymous reviewers for providing their constructive suggestions. Please find our response to the review comments below in italics.

Reviewer 2

According to my opinion this paper requires extensive revisions. Abstract and conclusions does not seem of standard profession level. Authors are advised to rewrite the abstract and conclusions sections.

Thank you for this consideration. We have carefully reviewed and rewritten the abstract and conclusions to remove any potential issues relative to the guidelines provided by MDPI Sensors, which are copied below. We believe all elements are represented in the revised abstract:

“The abstract should be a single paragraph and should follow the style of structured abstracts, but without headings: 1) Background: Place the question addressed in a broad context and highlight the purpose of the study; 2) Methods: Describe briefly the main methods or treatments applied. Include any relevant preregistration numbers, and species and strains of any animals used. 3) Results: Summarize the article's main findings; and 4) Conclusion: Indicate the main conclusions or interpretations. The abstract should be an objective representation of the article: it must not contain results which are not presented and substantiated in the main text and should not exaggerate the main conclusions.”

Also this paper contains a lot of unnecessary basic information which researchers already know, so focus on the key points more rather than providing the introductions.

This may be partially due to the article type; as we review the current state of the art, some introductory notes are provided to clearly set up the scope and context of the review, as well as provide definitions of key terms in the field. As a part of the review paper concerns a bibliometric / clustering analysis, some basic supporting theory regarding TF-IDF and k-means clustering is provided to ensure reproducibility of the results. We are open to suggestions as to specific elements of basic information which could or should be removed, but hesitate to make indiscriminate cuts as the review comment above can be interpreted in multiple ways.

It would be better if you use more Figs. to explain your work. It is not necessary to copy paste the diagrams from others, you can use your own block level diagrams after going through the related research material.

Our work contains six figures, of which four are original figures generated as part of this review, and five tables, all of which are generated as part of this review. The only exceptions are 1) Figure 5, which we’ve included to illustrate the US Department of Defence’s focus on transitioning to what they style as ‘CBM+’, with our explanation of this concept in the preceding paragraph, noting that the DoD’s support of this concept across multiple application domains makes it one of the strongest pushes for predictive maintenance in defence context and is therefore important for us to represent directly; 2) Figure 6, which is a heavily adapted version of a preceding figure from an MIT PhD thesis, with the core axis of the figure entirely new and based on this review.

In addition to these visual representations, we have included a variety of equations in our work. In text-heavy sections of the review (in particular Section 5, which discussed future challenges and opportunities), we employ bulleted lists to structure the narrative and break up the flow. However, given the variety of aspects discussed in this section, we did not see many opportunities to include figures or tables to accompany the discussion. However, as for the previous reviewer comment, we would definitely welcome suggestions for sections / concepts that would merit illustration and would be very happy to action this upon the reviewer directions.

Round 2

Reviewer 1 Report

The authors made all required adjustments and explanations exhaustively. 

Reviewer 2 Report

This can be accepted for publication